# Gotta Go Slow: Two Evolutionarily Distinct Annelids Retain a Common Hedgehog Pathway Composition, Outlining Its Pan-Bilaterian Core

**DOI:** 10.3390/ijms232214312

**Published:** 2022-11-18

**Authors:** Sofia Platova, Liudmila Poliushkevich, Milana Kulakova, Maksim Nesterenko, Viktor Starunov, Elena Novikova

**Affiliations:** 1Faculty of Biology, St. Petersburg State University, Saint Petersburg 199034, Russia; 2Zoological Institute RAS, Saint Petersburg 199034, Russia

**Keywords:** hedgehog signaling, pathway, transcriptome analysis, Spiralia, annelids, molecular evolution, *Platynereis dumerilii*, *Pygopsio elegans*

## Abstract

Hedgehog signaling is one of the key regulators of morphogenesis, cell differentiation, and regeneration. While the Hh pathway is present in all bilaterians, it has mainly been studied in model animals such as *Drosophila* and vertebrates. Despite the conservatism of its core components, mechanisms of signal transduction and additional components vary in Ecdysozoa and Deuterostomia. Vertebrates have multiple copies of the pathway members, which complicates signaling implementation, whereas model ecdysozoans appear to have lost some components due to fast evolution rates. To shed light on the ancestral state of Hh signaling, models from the third clade, Spiralia, are needed. In our research, we analyzed the transcriptomes of two spiralian animals, errantial annelid *Platynereis dumerilii* (Nereididae) and sedentarian annelid *Pygospio elegans* (Spionidae). We found that both annelids express almost all Hh pathway components present in *Drosophila* and mouse. We performed a phylogenetic analysis of the core pathway components and built multiple sequence alignments of the additional key members. Our results imply that the Hh pathway compositions of both annelids share more similarities with vertebrates than with the fruit fly. Possessing an almost complete set of single-copy Hh pathway members, lophotrochozoan signaling composition may reflect the ancestral features of all three bilaterian branches.

## 1. Introduction

The Hedgehog (Hh) pathway is found in all Metazoa except for Ctenophora, Placozoa, and Porifera, and the single components of this signaling pathway were identified in Protista [1,2,3,4]. Cnidaria seem to be the earliest group of living organisms to possess the full complement of Hh cascade participants [5,6]. 

The Hh pathway is one of the most important elements of cell–cell communication and the key regulator of many fundamental developmental processes on multiple levels in all bilaterian animals. Canonical Hh signaling can work as a switch between cell fates or function as a morphogen with short- or long-range action, controlling cell differentiation in the developing rudiments in a dose-dependent manner, for example, in the vertebrate limb buds or neural tube [7,8,9]. In addition, Hh signaling can act in a noncanonical way during cell differentiation and regeneration (reviewed in [10]).

The most detailed studies of the presence of the Hh signaling components and their functioning have been performed for model animals from Deuterostomia and Ecdysozoa: *Mus musculus* and *Drosophila melanogater* (for details, see the Results section) [11,12]. These studies revealed that the core components of Hh signaling—Hedgehog ligand, transporting protein Dispatched, Patched and Smoothened receptors, and transcriptional factor Gli/Ci (Glioma-associated oncogene/ Cubitus interruptus)—are structurally and functionally conserved in *Drosophila* and vertebrates [8,11,13] (Figure 1A,B).

While the key components of Hh signaling are conserved between model proto- and deuterostomes, there is remarkable variation in the number of core proteins and additional proteins, which supports necessary modifications of the active signaling components [8,15,16]. The mode of signal transduction and the cellular localization of Hh signaling also vary between mammals and *Drosophila* [8,13,15]. In mammals, Hh signaling occurs in the primary cilium, the antenna-like structure that protrudes from the cell surface [17]. In *Drosophila*, similar events take place on the cellular membrane, since the fruit fly’s cells lack cilia during development. Nevertheless, Hh signaling attached to the primary cilium has been found in *Drosophila* olfactory sensory neurons [18]. 

These counterpoints raise several questions: What was the ancestral composition of Hh signaling? How was the signal transduction realized in Urbilateria, and where it was localized in the cell? In order to answer these questions, research on the Hh signaling composition in nonmodel members of the third bilaterian clade, Spiralia, is needed. Current phylogenomic analyses imply that Spiralia and Ecdysozoa are sister taxa, while Deuterostomia is the basal group in Nephrozoa [19,20]. Therefore, similarities and differences in the Hh signaling compositions of Spiralia and Deuterostomia will allow ancestral traits and taxon-specific changes in each Nephrozoa lineage to be distinguished.

Among Spiralia, the expression of Hh signaling components has been examined in annelids, molluscs, and planarians. In annelids, Hh signaling has been shown to participate in larval development, regeneration, and adult growth [21,22,23,24,25]. In molluscs, a distinct Hh-related family named Lophohog has been identified, and Hh has been shown to be involved in cephalopod limb and muscle development [26,27,28,29]. *Hh* expression in brachiopods is restricted to embryonic endodermal tissues [30]. In the planarians *Schmidtea mediterranea* and *Dugesia japonica*, Hh signaling is responsible for the establishment of anterior–posterior polarity [31]. Moreover, investigating planarian regeneration, Rink and co-authors (2009) attempted to unravel the evolutionary connection between Hh signaling and the function of cilia [32]. These studies have been conducted exclusively on the core components of Hh signaling, which are conserved among Nephrozoa, and we did not manage to find a full-size search of the whole signaling repertoire for any spiralian animal. To clarify the evolution of the Hh signaling in Bilateria, it is necessary to study the spiralian composition of noncore proteins, which are not equivalent in *Drosophila* and vertebrates. 

Annelids are the perfect models for studying the functional and structural evolution of genes which control morphogenesis [33]. They are efficiently accessible for use of molecular techniques, and their ancestral character is reflected at both the gene organization and gene expression levels [34,35,36]. Multiple evolutionary questions that arise while studying vertebrate and arthropod models are related to the search for the basal regulatory states of various processes. For example, vertebrates and ecdysozoans differ in their compositions of neuropeptides and neuromediators. In spiralian animals (e.g., *Platynereis*) both sets of these molecules have been found [37], which clarifies the question concerning their ancestral repertoire. A similar situation emerges while studying the evolution of signaling pathways (RA, Hh, Wnt) or the functioning of transcription factors [21,22,38,39,40,41]. Thus, annelids are very promising models for studying the evolution of developmental molecular determinants. 

The present research focused on studying Hh signaling in two evolutionarily distant annelids: *Platynereis dumerilii* and *Pygospio elegans*. These sea worms belong to two distinct monophyletic evolutionary branches: Errantia and Sedentaria, respectively. These two branches split from the common ancestor around 500 mya. *Platynereis dumerilii* (Nereididae) is a homonomously segmented worm which inhabits the Mediterranean Sea, but can be easily cultivated in laboratory settings [42]. *Pygospio elegans* (Spionidae) demonstrates many features that can be regarded as ancestral for Sedentaria, such as small parapodia lacking internalized supporting chaetae, the absence of antennae, sedentary lifestyle, microphagy, and grooved (ciliated) palps [43,44]. It can be found throughout the boreal sea waters, but the worms used in this work were collected from the Barents Sea.

In this work, we analyzed the transcriptomes of *Platynereis dumerilii* and *Pygospio elegans* for the presence of components of Hh signaling. For *P. elegans*, we used transcriptomes from juvenile and regenerating worms ([45], manuscript in preparation). For *P. dumerilii*, we analyzed transcriptomes from worms at various stages of regeneration (manuscript in preparation). The assembled sequences of the *Platynereis* genes of interest were mapped onto the *Platynereis* genome sequences used as controls (genome data provided by K. N. Mutemi). In our analysis, we included only the components of the canonical Hh pathway, since they have been better studied to date. We made an effort to focus on molecular phylogeny and on the search for the ancestral Hh signaling state, which could be a characteristic of the common bilaterian ancestor. To date, this work is the first detailed search for the presence of the whole set of Hh signaling components in two spiralian animals.

## 2. Results

In the transcriptomes of *Platynereis dumerilii* and *Pygospio elegans*, we found 71 genes that encode known members of the Hh signaling pathway. The identified genes are presented in Table 1, which compares the signaling compositions in the two annelids with those of *Drosophila*, mouse, and human.

### 2.1. Hedgehog Ligand: Processing and Transport

The Hh ligand is synthesized as a precursor of 45 kDa and is processed into an active 19 kDa signaling peptide [46,47,48]. The precursor molecule bears the N-terminal signaling domain Hedge (Hh-N) and the C-terminal domain Hog (Hh-C), bound by a highly conserved site of autoproteolysis comprising the three amino acid residues GCF (Figure 2A) [1,6,26]. Hog consists of the Hint domain (Hedgehog intein), probably related to inteins, and the sterol-recognition region (SRR), which noncovalently binds to membrane cholesterol [49,50]. The autoprocessing of the precursor occurs on the membranes of the cytoplasmic reticulum through two consequent nucleophilic attacks of the carbonyl group of a glycine residue in the GCF site. The cleavage of Hh-N is followed by the covalent attachment of cholesterol to the glycine residue on the C-terminus of the Hedge domain [6,13,50,51,52]. The N-terminus of the Hedge domain is modified by a palmitate residue by means of the acyltransferase Skinny Hedgehog/Rasp (HHAT in mammals) [1,6,26,52]. Lipidic modifications play key roles in ligand secretion and mobility [7]. The mechanisms of Hh ligand processing and secretion are conserved among different animals [53].

It is assumed that the Hedgehog gene appeared in cnidarians through the merger of Hedge and Hog domain sequences and was retained throughout the evolution of most Planulozoa [1,5,6,26,54]. Two *hedgehog* (*hh*) genes have been found in *Nematostella vectensis*, but the last common ancestor of bilaterians probably possessed one *hh* gene, since most protostomes and invertebrate deuterostomes have only one *hh* ortholog in their genomes. Therefore, additional *hh* genes might be the result of lineage-specific duplication events [55]. Thus, mammals have three homologs—*sonic hedgehog* (*SHH*), *indian hedgehog* (*IHH*), and *desert hedgehog* (*DHH*); *Danio rerio* possesses five—*Dhh, Ihha, Ihhb, Shha*, *and Shhb*; and the lamprey has two—*Hha* and *Hhb* [53,55,56]. Their origins are considered to be the result of whole-genome duplications, and it is known that mammalian *HH* paralogs play different roles [53,57]. In contrast, some species have either lost the true *hh* gene or acquired additional proteins with Hog domains, or even done both. For instance, the *Hh* gene has been lost in *Caenorhabditis elegans* [1,26]. Instead of true Hh proteins, this nematode possesses proteins which contain the Hog domain conjugated with other secreted N-terminal domains: Wart-hog, Qua-hog, and Ground-hog [26]. Furthermore, in spiralian animals from Lophotrochozoa, the gene family Lophohog was found, members of which contain a Hog domain and a unique N-terminal domain. *Lophohog* genes have been found in the molluscs *Lottia gigantea* [26], *Lottia* cf. *kogamogai*, and *Acanthochitona crinita* [27], and in the annelid *Capitella* sp. I [26]. 

We found one copy of *Hh* each in the transcriptomes of *P. dumerilii* and *P. elegans*. The amino acid sequence length of Pdum-Hh is 424 a.a. and that of Pele-Hh—477 a.a. Multiple sequence alignments of Hedgehog proteins from different animal species demonstrated the high conservatism of Hedge and Hog domains (Figure 2A).

To establish potential evolutionary relationships between bilaterian Hh proteins, we performed Bayesian phylogenetic analysis. Analysis retrieved the (Hh, (Dhh, (Ihh, Shh))) topology, in agreement with previous works [7,58,59]. Bilaterian Hh proteins segregated by lineage into Spiralia, Deuterostomia, and Ecdysozoa, forming corresponding monophyletic clades (Figure 2B). Hh proteins from deuterostomes formed a basal clade, while Spiralians were a weakly supported sister group to Ecdysozoa. Inside Spiralia, Pdum-Hh was associated with Hh from *Perinereis*, another nereid polychaete worm, whereas Pele-Hh fell into one clade with *Capitella* Hh. While the analysis failed to resolve the evolutionary relationships between Hh proteins from annelids and molluscs, it was clear they fell into one clade. The observed topology and the absence of a second Hh protein in the studied spiralians and most other bilaterians support the hypothesis that the common bilaterian ancestor possessed one Hh gene.

In the transcriptomes of the studied annelids, we did not find *Lophohog* genes (the Hedgehog-related genes which are specific for lophotrochozoans). We also could not reveal the presence of any other genes bearing the Hog/Hint domain except for Hedgehog sequences. 

*Skinny/HHAT* orthologs are not found in protists or sponges, while vertebrates, *Drosophila*, and *Nematostella* all possess one copy [5,60]. Both annelids have homologs of the *HHAT* gene; notably, we found two paralogs in the transcriptome of *P. dumerilii*, *Pdum-Hhat1* and *Pdum-Hhat2* (53% similarity for protein sequence) (Appendix A). Both amino acid sequences of Pdum-Hhat and the sequence of Pele-Hhat have an MBOAT domain and twelve transmembrane helices (Figure 3). 

Modified Hh-N leaves the cell with the help of Dispatched and Scube2 proteins [12,13,51,62]. Dispatched is structurally close to the Patched receptor, described further below, and belongs to the RND (Resistance-Nodulation-Division) family of transporters. Its activity releases Hh-N out of the cell [52]. Hh interacts with Dispatched via its cholesterol anchor and is transported to the membrane surface, where it interacts with lipid-binding Scube2 protein, which is required for the release and mobilization of Hh-N [12,14]. Upon being secreted out of the cell, Hh-N enters the extracellular space and can reach the target cell [12,13,51,52]. The effective transport of Hh-N depends on heparan sulfate proteoglycans (HSPGs). The long-distance transportation of the ligand is realized in complexes with lipoproteins or exosomes [7,12,13,14,53]. 

*Disp* is an evolutionarily ancient gene. Sequences homologous to *Disp* have been found in the genomes of both *Monosiga brevicollis* and Homoscleromorpha sponges [60,63]. In the transcriptomes of *P. dumerilii* and *P. elegans*, five and six proteins, respectively, from the Dispatched/Che-14 family were found [64]. To establish which of them are orthologs of *Drosophila* and murine DISPs, we constructed a phylogenetic tree based on whole amino acid sequences of DISP, PTC, and NPC1 proteins (Appendix A). Two of the eleven annelid proteins were associated with the Disp/Disp1/Disp2 clade. We named them Pdum-Disp1 and Pele-Disp1 and used them for further analysis. The remaining nine proteins formed two distinct clades; one clade was in a sister relationship to the Disp/Disp1/Disp2 clade, and the other formed a sister branch to metazoan Disp3 proteins. We did not find putative Disp3 orthologs in either of the studied polychaetes. 

The amino acid sequences of Pdum-Disp1 and Pele-Disp1 comprise 1137 and 1194 a. a., respectively. Both proteins have twelve transmembrane helices and conserved Patched and sterol-sensing domains (Figure 4A). Phylogenetic analysis of known bilaterian DISPs shows that proteins separate into two clades: the Disp1/Disp2 clade and the DISP3 clade (Figure 4B). Pele-Disp1 and Pdum-Disp1, together with other Dispatched proteins which have been shown to participate in Hh signaling, fell into the Disp1/Disp2 clade. Inside the Disp1/Disp2 clade, invertebrate species possessed only one form of Disp, while vertebrates had two DISP copies, suggesting that DISP1 and DISP2 proteins may have arisen via gene duplication within vertebrate lineage. Our phylogenetic analysis supported a sister relationship between ecdysozoan and spiralian Disps, while deuterostomes were at a more basal position inside the Disp1/Disp2 clade. Therefore, Pele-Disp1 and Pdum-Disp1 may secrete the Hh ligand in the same manner as *Drosophila* Disp. 

The Scube family is best studied in vertebrate species and it consists of the secreted proteins Scube1, Scube2, and Scube3. Of these, only Scube2 is involved in long-range Hh signaling [65]. In the transcriptomes of *P. dumerilii* and *P. elegans*, we found single homologs of the Scube proteins: Pdum-Scube and Pele-Scube (Appendix A). 

### 2.2. Hh Receptor Patched (Ptc), Co-Receptors, and Antagonists

The Hh ligand interacts with its twelve-pass transmembrane receptor Patched (Ptch, Ptc), which possesses the specific Patched domain and sterol-sensing domain (SSD). Ptc belongs to the RND (Resistance-Nodulation-Division) family of transmembrane transporters and demonstrates similarity to Niemann–Pick C1 protein, the regulator of cholesterol exchange which also bears the SSD domain [7,51,52,66,67]. The role of Patched is the suppression of another transmembrane protein, Smoothened (Smo), in the absence of the Hh signal. After the Hh ligand interacts with Ptc, the suppression is removed and the activated Smo transmits the Hh signal inside the cell [67,68]. 

Twelve hydrophobic transmembrane domains of Ptc are conserved among proto- and deuterostomes [8,15,51,67,69]. Two homologs of *Ptc* (*Ptch1* and *Ptch2)* have been found in vertebrates (mammals and *Danio rerio*), and their functions mainly overlap. Only one *Ptc* gene has been found in invertebrates [8,12,52]. 

Besides its main receptor, the Hh ligand interacts with other cytosolic membrane proteins—the co-receptors Ihog/CDON, Boi/BOC, and GAS1—which enhance the association of the ligand with the main receptor. Moreover, Hh interacts with some proteins of the HHIP family. It is known that HHIP1 acts exclusively as the pathway antagonist through interaction with the ligand for its subsequent internalization [1,52,66]. It is noteworthy that the presence of these pathway components varies among different groups. For example, GAS1 is present in vertebrates, nematodes, and crustaceans, but is absent in *D. melanogaster* [70]. The transmembrane protein HHIP is specific to vertebrates and is not found in *D. melanogaster*. Meanwhile, three main metazoan groups, including Cnidaria, possess *HHIP-like* genes [1,5]. It is known that HHIP2 (HHIP-like 1) positively regulates Hh signaling, as shown by experiments on human aortic smooth muscle cells [71], but its participation in the regulation of Hh signaling in other metazoans was not shown. Ihog/CDON and Boi/BOC proteins participate in the Hh signaling of vertebrates and *D. melanogaster* [1]. Moreover, the binding mode of Hedgehog with Ihog/CDON homologs differs between *Drosophila* and mammals [72]. 

In the transcriptomes of both annelids, single homologs of *Patched*, *Pdum-Ptc* and *Pele-Ptc*, were found (Figure 5A). The amino acid sequences of Pdum-Ptc and Pele-Ptc comprise 1353 and 1529 a.a., respectively, and contain the Patched domain (PFAM number: PF02460). Both proteins have twelve conserved transmembrane helices (TM), five of which form the sterol-sensing domain (SSD). Moreover, according to DeepTMHMM results, Pdum-Ptc and Pele-Ptc possess two extracellular domains (ECD-I and ECD-II). It was previously shown that extracellular domains (ECD-I and ECD-II) are necessary for human PATCHED Hh-N ligand binding [73]. Based on the domain compositions of Pdum-Ptc and Pele-Ptc, these proteins are functional receptors of the Hh ligand. 

To establish evolutionary correspondence between Pdum-Ptc, Pele-Ptc, and the other metazoan Patched proteins, we subjected predicted protein sequences to Bayesian phylogenetic analysis (Figure 5B). Although Ptc proteins segregated into monophyletic lineages, the phylogenetic analysis failed to resolve relationships between the Spiralia, Ecdysozoa, and Deuterostomia clades. Pdum-Ptc and Pele-Ptc, together with Ptc from brachiopods and molluscs, clustered into the Spiralia clade. Pele-Ptc united with *Capitella* Ptc into a Sedentaria branch; Pdum-Ptc was a sister taxon to this branch.

In the *P. dumerilii* and *P. elegans* transcriptomes, we found the sequences of *Gas1* (Appendix A) and *HHIP-like* (Appendix A); orthologs of *Hhip* were not found. It is noteworthy that in the annelids’ transcriptomes, a sequence was found which cannot be referred to as an ortholog of Ihog/CDON, nor one of Boi/BOC, since it is equally similar to both of them (Appendix A). We called this homolog of *CDON/BOC* and *Ihog/Boi Fibbc* (Father of Ihog/Boi/BOC/CDON) (Appendix A).

### 2.3. Smoothened 

The most important participant in the Hh signaling pathway is the transmembrane receptor Smoothened (Smo), which forms dimers and transmits the Hh signal from the surface of the recipient cell to its cytoplasmic components [74,75]. Smo consists of a highly conserved, extracellular N-terminal cysteine-rich domain (FRI/CRD), heptahelical transmembrane domain (TMD/Frizzled), and a long intracellular tail in the C-terminal domain (CTD) [8,69,76]. Smo is a member of the G-protein-coupled receptor (GPCR) family, and it displays significant similarity to the Frizzled protein (F-class GPCR), which is the receptor of the Wnt pathway [76,77].

In the canonical signaling pathway, Smo does not directly interact with the Hh ligand. Its state is regulated by the Ptc protein through secondary messengers, and cholesterol and oxysterol seem to be the most suitable for this role [57,68,78]. Active Ptc protein (which is not inhibited by the Hh ligand) reduces the available cholesterol in the cellular membrane and maintains Smo in its inactive conformation in the endosomes. The moment Ptc is inhibited by the Hh ligand, the level of available cholesterol increases and the activated Smo is transported to the cell membrane. The concentration of Smo dimers on the cell surface, the conformational availability of their intercellular domains for phosphorylation, and the extent of this phosphorylation define the intensity of signal transmission inside the cell [68,79]. In other words, differential response to the Hh ligand depends not only on its physical gradient but also on the condition of the Smo receptors. 

Interestingly, the long intracellular tail in the C-terminus differs between vertebrates and *Drosophila* [8]. These differences are responsible for the lineage-specific regulation of Smo proteins. For instance, the set of protein kinases which phosphorylate Smo intracellular domains do not fully overlap for mouse and fruit fly. It is known that the Smo of *D. melanogaster* is phosphorylated by protein kinase A (PKA), casein kinase 1 (CK1), CK2, and G-protein-coupled receptor kinase 2 (Gprk2/GRK2) [14,75,80]. Meanwhile, for vertebrates, these kinases are CK1 and GRK2 [14,75] (Figure 6A). Moreover, *Drosophila* and vertebrate Smo proteins form complexes with distinct components of the Hh signaling pathway. In *Drosophila*, Smo physically interacts with Cos2 through the interaction domain in the C-terminus, and this domain is not conserved in the zebrafish Cos2 homolog Kif7 [81,82,83]. In vertebrates, the C-terminus of Smo interacts with several proteins of the BBSome complex, which are involved in transporting Smo to cilia [84]. 

The Smo gene and similar sequences are not found in protist genomes or in the genome of ctenophore *Mnemiopsis*, but they have been described for sponges from Calcarea, Hexactinellida, and Homoscleromorpha [60,63,85]. Despite the variations in the sequence of the intracellular tail, Smo is a highly conserved component of the Hh pathway. It is present as a single copy in the genomes of proto- and deuterostomes. 

Single *Smo* genes were found in the transcriptomes of *P. dumerilii* and *P. elegans.* The amino acid sequences of Pdum-Smo and Pele-Smo comprise 1005 a.a. and 1404 a.a., respectively. Both proteins contain the conserved FRI/CRD and Frizzled domains, the conserved transmembrane regions of seven TM helices, an extracellular N-terminus, and an intercellular C-terminus (Figure 6B). 

It was found that Pele-Smo possesses a long C-terminal domain of 828 a.a., and the length of the Pdum-Smo CTD is 351 a.a. Multiple alignment with Dm-Smo did not reveal any significant similarities in the CTD structures of fruit fly and the studied worms (21.5% similarity *Pygospio* vs. fruit fly; 30.6% similarity *Platynereis* vs. fruit fly) (Figure 6B). Similar results were obtained for the comparison of studied worm and murine CTDs (15.3% similarity *Pygospio* vs. mouse; 25.3% similarity *Platynereis* vs. mouse). Moreover, the CTD domains of the worms demonstrated a very faint resemblance to each other (20% identical, 30.6% similar).

Although the CTD is the most divergent region of the protein, it is essential for Smo activity, as it contains a set of sites for phosphorylation by several protein kinases and regions for interaction with other signaling components. In the Pele-Smo C-terminal tail, there are 61 serine and 10 threonine sites predicted to be phosphorylated by PKA, CK1, CK2, and GRK2 (Figure 6A). In the Pdum-Smo C-terminal tail, there are 33 serine and 7 threonine sites predicted to be phosphorylated by the same set of kinases. Furthermore, in the studied spiralians, we detected at least three conserved PKA phosphorylation sites that are not present in *Drosophila*. The Pele-Smo and Pdum-Smo regions that align against *Drosophila* Cos2-binding domains do not share considerable sequence homology with the latter (5–19% identity and 17–33% similarity with the first Cos2-binding domain; 8–9% identity and 16–20% similarity with the second Cos2-binding domain). Moreover, the sequences in Pele-Smo and Pdum-Smo which correspond to the vertebrate BBSome interaction region do not demonstrate enough similarity (60–63% similarity) to speculate that both proteins might interact with the BBSome. These observed differences in sets of phosphorylation sites and protein interaction regions support the argument that spiralian Smo proteins are regulated differently from those of vertebrates and *Drosophila*. 

Due to significant divergence in the CTD, we subjected more conserved FRI/CRD and Frizzled domains to Bayesian phylogenetic analysis (Figure 7). The phylogenetic analysis results support the identification of an Annelida clade composed of Pele-Smo, Pdum-Smo, and *Capitella* Smo. Smo from another polychaete worm, *Owenia*, did not fall into the Annelida clade, and a sister relationship between *Owenia* Smo and brachiopod Smo was weakly supported. 

Spiralian Smo proteins formed a monophyletic clade, sister to Deuterostomia and *Nematostella* Smo proteins, while ecdysozoan Smo proteins formed a basal branch. This suggests that the N-terminal regions of Pele-Smo and Pdum-Smo may have similar functions to the vertebrate FRI/CRD and Frizzled domains.

### 2.4. Intracellular Components of the Hh Pathway

Activation or nonactivation of the Hh pathway alters the condition of the transcriptional effector protein Gli/Ci. In the absence of the Hh signal, this protein is always phosphorylated, which leads to its partial proteolysis and domination of the repressor form Gli-R. After activation of the Hh cascade, the phosphorylation is disrupted and the full-length Gli/Ci, Gli-A, works as an activator. The mechanisms by which the signal from Smo transduces to the effector are not fully clear. The main participants in this transduction partially overlap between mammals and *Drosophila*, but the differences are essential enough to consider this part of the signaling to be the least conserved.

It is known that in *Drosophila*, the intracellular components of the Hh pathway are associated with the microtubules and are presented by the so-called HSC (Hedgehog Signaling Complex), which includes Ci, Costal-2 (Cos2) protein, Ser/Thr kinase Fused (Fu), Suppressor of Fused (SuFu) protein, and PKA, CK1, and GSK3b kinases [15]. Cos2 is a kinesin-like protein and is the most important component of Hh signaling in *Drosophila*, since it functions as a scaffold which recruits the remaining cytoplasmic components of signal transduction. In the absence of the signal, Cos2 connected to microtubules interacts with Fu, SuFu, and the Cos2-binding domain (CORD) of the transcriptional factor Ci [67,81,82,86,87,88]. Being connected to Cos2, the Ci protein is phosphorylated by the combined action of PKA, CK1, and GSK3b kinases. 

Phosphorylated Ci is ubiquitinated and directed to proteasomes for partial proteolysis [67,75,88]. The switch of phosphorylation substrate for PKA probably occurs from Ci to Smo CTD after Hh interacts with Ptc [89]. Smo, which is also phosphorylated by CK1, CK2, and GPRK2 kinases, transits into its active state and interacts with HSC due to the binding to Cos2 [75]. Interaction of Cos2 with the Smo CTD leads to the release of Ci, which stops its phosphorylation by kinases and prevents cleavage. The activated Fused is a positive regulator of this signaling, which alleviates the dissociation of Ci from SuFu (Figure 1A) [82,83,88]. 

Although the Smo CTD domain and Cos2 are important participants in Hh signal transduction in *Drosophila*, these components do not play any essential role in the signaling in vertebrates [8,16,57]. The pivotal feature of vertebrate Hh signaling is the dependence of the pathway on the primary (sensory/nonmotile) cilium [9,90,91,92,93,94]. The cilia function as cellular compartments in which signal transduction occurs (Hh, Wnt, Notch) via trafficking of signaling components [95,96]. 

In mammals, two *Cos2* orthologs have been found, *KIF7* and *KIF27*, as well as one ortholog of *SuFu* and one ortholog of *Fused*, *STK36/FU*. It has been shown that KIF27 and FU proteins are not essential for signal transduction in mammals and participate in motile ciliogenesis [92,97]. Strikingly, zebrafish Fu and KIF7—the only identified Cos2 ortholog in zebrafish—participate in both motile ciliogenesis and Hh signaling [97].

Nevertheless, SUFU and KIF7 proteins are necessary for Hh signaling in mammals [92,97,98]. The primary function of KIF7 is probably the alignment of plus-end microtubules at the tip of the cilium via a concentration of GLI with which it can interact [98,99]. The localization of GLI at the end of the cilium is a crucial step in mammalian Hh signaling, since the main signaling components are concentrated in this compound [98,99,100]. It is interesting that in contrast to Cos2, KIF7 and KIF27 do not interact with SMO. In contrast, SUFU binds to GLI in a conserved way and plays an evolutionarily conserved role in promoting GLI2/3 stabilization. Together, they are concentrated on the tip of the cilium [97]. 

In the absence of Hh signaling, the KIF7-SUFU-GLI complex is primarily localized at the base of the primary cilium [92]. Where it happens, the full-sized GLI is phosphorylated by PKA, GSK3b and CK1 kinases, which leads to its cleavage to the repressor form [75]. After ligand binding to PTC, SMO activates and moves to the membrane of the primary cilium [11,101]. Upon Hh pathway activation, KIF7-SUFU-GLI accumulates in the primary cilium tip. The activated SMO mediates GLI release and prevents its phosphorylation and proteolysis [9,92]. There are also many additional regulators of Hh signaling that act in mammals (Figure 1B; Table 1) [9,92,102].

In the transcriptomes of *P. dumerilii* and *P. elegans*, 62 genes encoding the components of intracellular Hh signaling were found (Table 1). Out of the genes which encode proteins from the HSC, we found homologs of *Fu/Stk36*, *SuFu*, and *Kif27* in the annelids’ transcriptomes, but the *Kif7* sequence was not found (Appendix A and Figure 8). We also found multiple genes associated with the primary cilium. These genes are presented in Table 1 and their sequences are given in Appendix A.

### 2.5. Gli/Ci: The Transcriptional Effector of Hh Signaling

The terminal participant of the Hh signaling cascade is the Gli/Ci protein, a bifunctional transcription factor which has the repressor N-terminal and the activator C-terminal domains (Figure 1A,B). These domains flank the central part of the protein, which consists of five DNA-binding ZnF_C2H2 domains [13,67]. The regulation of processing and the nuclear translocation of Gli/Ci proteins play key roles in the Hh signaling pathway. The balance between the activator and the repressor forms of the transcription factor in a cell depends on the number of ligand molecules and the efficacy of signal transduction upstream of Gli/Ci. Due to the gradient distribution of Hh molecules and the readiness of the cell to interpret the signal and transmit it to the nucleus, a continuum of regulatory states, which can cause variable cell responses during morphogenesis or for maintenance of tissue homeostasis, is established [7]. When the concentration of the Hh ligand is low, the repressor form of Gli/Ci dominates in the cell due to phosphorylation of the C-terminus and to recognition by Slimb/β-TrCP proteins, which direct it to the proteasome for partial degradation of the C-terminal domain. High concentration levels of the ligand inhibit the proteolysis of Gli/Ci, and the full protein form accumulates in the cell and acts as a transcription activator [11,67,91]. Homologs of Gli/Ci are not found in Protista, but they are present in all representatives of Porifera studied to date [60]. In cnidarian and protostomian genomes, a single *Gli/Ci* gene is present, while mammals possess three homologs—*GLI1*, *GLI2*, and *GLI3.* GLI1 is not processed to the repressor form and works as an activator [7,67,103].

In the transcriptomes of both annelids, only one homolog of *Gli* was found. Pdum-Gli and Pele-Gli comprise 1824 and 1557 amino acids, respectively, and bear the conserved regions of five ZnF_C2H2 DNA-binding domains in the central part of the sequence (Figure 9). On a phylogenetic tree, both proteins formed a distinct branch within the monophyletic Spiralia clade (Figure 10). Spiralian Gli proteins were in a sister relationship with ecdysozoan Ci proteins, while deuterostomes formed a basal clade. In deuterostomes, the vertebrate GLI proteins formed a monophyletic clade, within which GLI2 and GLI3 proteins clustered together, whereas GLI1 proteins formed a sister group to them, in agreement with earlier analyses [103]. 

## 3. Discussion 

In our study, we used nonclassical models for the investigation of Hh signaling: annelids *Platynereis dumerilii* and *Pygospio elegans*. In these two phylogenetically distant sea worms, we detected almost full complements of Hh-pathway-related genes, which shared more similarities with the profiles of vertebrates than those of model ecdysozoans (Figure 11). Although Spiralia and Ecdysozoa are sister taxa, the number of unique signaling components in these annelids differs from that of fruit flies by 30% (Table 1). Additional differences in the Hh signaling composition among Bilateria are discussed below. 

### 3.1. The Main Core Components: Phylogeny and Protein Composition

The highly conserved core of the signaling pathway, represented by the Hh ligand, Patched and Smo receptors, and transcriptional factor Gli/Ci, was found in both annelids’ transcriptomes. Similarly to the results of previous studies of protostomes and also echinoderms [5,21,25,30,69,104,105], one ortholog of each main component of the signaling was found in each annelid transcriptome. One more conserved component of the Hh cascade is the transmembrane transporter Dispatched. In the *P. elegans* and *P. dumerilii* transcriptomes, we found several sequences homologous to the Dispatched/Che-14 family genes, but according to the results of the phylogenetic analysis, only one sequence can be considered with certainty to be a homolog of *Drosophila Disp* and murine *Disp* (Figure 4B).

The results of the phylogenetic analysis show that Hh, Ptc, Smo, Gli, and Disp clearly segregate according to the main metazoan lineages: Deuterostomia, Spiralia, and Ecdyzozoa. All Bayesian trees, except trees for Smo and Kif, exhibited topologies consistent with current understandings of bilaterian evolutionary relationships [19,20]. That is, proteins from deuterostomes formed basal clades in relation to spiralian and ecdysozoan sister clades. The observed topologies support a monophyletic origin of each gene, whereas the absence of corresponding paralogs in most bilaterians implies that the common bilaterian ancestor only possessed one copy of each core gene. A similar conclusion was drawn by Matus et al. 2008 in a study on the Hh pathway in *Nematostella vectensis*, in which it was suggested that the cnidarian–bilaterian ancestor possessed one Hh gene [5]. Here, we extend this hypothesis to all core components of the Hh pathway, suggesting that formerly single-copy genes underwent duplication events within several bilaterian lineages. Due to independent duplications, *N. vectensis* and *C. intestinalis* possess two Hh genes, while in vertebrates, whole-genome duplications gave rise to paralogs of all core proteins. 

In the studied transcriptomes, we did not find Hint-only genes or any other sequences with the Hog/Hint domains, including the Spiralia-specific Hh homolog Lophohog. In previous studies, Lophohog was found in the molluscs *Lottia gigantea*, *Lottia* cf. *kogamogai*, and *Acanthochitona crinita*, and in the annelid *Capitella* sp. I [26,27]. At the same time, the *Lophohog* gene was not detected in other mollusc species (*G. pellucida*, *W. argentea*, *S. ventrolineatus*, *I. notoides*, *N. tumidula*, and *A. entalis*) that were studied by De Oliveira and colleagues. The authors mention that the reason may lie in sequencing peculiarities and might not reflect the true picture [27]. It is likely that we could not find *Lophohog* in our annelids because it is not expressed at the studied time points. It is noteworthy that the structural evolution of the Hh ligand correlates with the fast evolution rate and divergence of the Hh pathway. In particular, three genes with the Hog domain have been found in nematodes [26]. To clarify the question of Lophohog ancestrality, further genomic and transcriptomic studies are needed. 

The detailed analysis of amino acid sequences in domains of the core signaling components (Hh, Disp, Ptc, Smo, Gli) and their multiple alignments demonstrated high conservatism of their functional domains. Nevertheless, the structure of the Smo proteins of the studied annelids demands special mention. The Smo CTD is considerably divergent between vertebrates and *Drosophila*, and it is assumed that this variability lies behind the differences in the mechanisms of signal transduction. Vertebrate Smo does not possess a long C-terminal domain and does not bear the Cos2-protein-binding site. The ~180 amino acids of CTD located closest to the transmembrane domain are relatively conserved [8,57]. The PKA phosphorylation sites are not found in the vertebrate SMO CTD; SMO activation in vertebrates depends on CTD phosphorylation by CK1 and GRK2 kinases [75]. 

Transcriptome analysis of *P. dumerilii* and *P. elegans* allowed us to find the full sequences of Pdum-Smo and Pele-Smo, which are significantly divergent in their C-termini. The functional domains of both sequences were analyzed and included in our phylogenetic analysis. Our multiple alignment of the amino acid sequences of different protostome, deuterostome, and annelid Smo proteins supported the conservatism of the extracellular N-terminal and seven TM helix domains, while the CTD domain was varied in our alignment. The *Drosophila* Smo CTD is of considerable length and includes two sites which are critically important for Cos2 binding (between amino acids 652–686, and 730–1035) [16,81]. However, the multiple alignment of Pdum-Smo, Pele-Smo, and Dme-Smo did not reveal any similarities in the structures of the C-terminal domains. The Cos2-binding site was not found in the annelids’ Smo sequences. According to previously published data, the Cos2-binding site is retained in insects and crustaceans, but is not found in vertebrates, sea urchins, annelids, or spiders [1,104]. Despite this, Pdum-Smo and Pele-Smo CTD analysis showed the presence of phosphorylation sites for PKA, CK1, CK2, and GRK2 kinases. Only a few sites overlapped between the annelids, mouse, and *Drosophila*. In the studied spiralians, we detected four conserved PKA phosphorylation sites, in agreement with previous work conducted on the mollusc *Crassostrea gigas* [69]. However, three of these sites are not present in *Drosophila*, which suggests that spiralian Smo proteins may be regulated differently. 

The opposite part of the Smo protein, containing the N-terminal extracellular cysteine-rich and Frizzled domains, is relatively conserved across metazoans. Multiple sequence alignment revealed that the corresponding Pele-Smo and Pdum-Smo regions are more similar to those of murine Smo than of *Drosophila* Smo. It has been shown that the transmembrane domain of mammalian Smo is sensitive to exogenous small molecules (such as alkaloid cyclopamine), but this has not been observed in *Drosophila* [8,76,106,107]. It is noteworthy that annelid and mollusc Smo proteins are also sensitive to cyclopamine action [22,23,28,29]. This similarity is in agreement with the results of our phylogenetic analysis—on the tree, deuterostomian and spiralian Smo proteins fell into one clade, while ecdysozoan Smo branched earlier and formed a separate sister clade. 

### 3.2. Hedgehog Ligand Transport Outside the Cell

The presence of the Intein N-terminal splicing motif in the Hog domain of Pdum-Hh and Pele-Hh, as well as the presence of acetyltransferase *Hhat* in both annelids’ transcriptomes, may point to retention of the evolutionarily conserved mechanism of Hh ligand processing in the annelid lineage. Note that the two paralogs of *Hhat* found in the *P. dumerilii* transcriptome have matching functional domains (Figure 3). This is probably the result of clade-specific duplication. 

The mechanism of secretion of modified Hh-N out of the cell by the Dispatched membrane transporter is probably retained in annelids, as judged by the conserved domain organization of Pdum-Disp1 and Pele-Disp1 (Figure 4A). The Disp1 sequences from the studied annelids clustered with Disp1 from the other nephrozoans, which points to its high conservatism. The phylogenetic position of the protein sequences corresponds to the modern understanding of the relationships between taxa inside the Spiralian group (Figure 4B). 

In the transcriptomes of both annelids, single Scube homologs were found (Appendix A). However, the results of phylogenetic and domain composition analyses did not allow any conclusion to be drawn concerning the involvement of Pele-Scube and Pdum-Scube in signaling. 

Thus, the mechanism of Hh-N signal molecule release from the producing cell in annelids may be similar to that in either *Drosophila* or vertebrates, but this cannot be determined without functional tests.

### 3.3. Hedgehog Receptor Binding

The presence of membrane proteins Ihog/CDON, Boi/BOC, HHIP, and GAS1 and their participation in Hh signaling vary among different animal groups. In the transcriptomes of both worms, we found a single sequence of the *Gas1* gene, which aligned well with the orthologous genes of other bilaterians (Appendix A). The presence of the *Gas1* gene in both vertebrates and spiralians (i.e., annelids) and also in cnidarians (according to NCBI), while it is absent in *Drosophila* is most likely the result of a common origin of this gene in Metazoa. 

We did not find *Hhip* genes in the studied annelids, but there were *Hhip-like* genes in both transcriptomes (Appendix A). It is known that their orthologs do not participate in Hh signaling, except for a single case described for human aortic smooth muscle cells [71]. The annelids’ *Hhip-like* genes formed a sister branch to the HHIP-like genes of vertebrates (Appendix A) and we cannot confirm their involvement in the Hh pathway. 

In the transcriptomes of both annelids, a single homologous gene was found with a sequence equally similar to those of Ihog/CDON and Boi/BOC (Appendix A). We named this gene *Fibbc* (Father of Ihog/Boi/BOC/CDON). The amino acid sequences of Pdum-Fibbc and Pele-Fibbc clustered well with the homologs from other spiralian animals and did not form a clade with those of drosophilids (Ihog and Boi) or vertebrates (CDON and BOC). 

It is noteworthy that model animals from Ecdysozoa and Deuterostomia also displayed the taxon-specific clustering of orthologs of these genes, which is in line with the results from Lencer and co-authors (2022) [108]. 

We suppose that the ancestral sequence of Ihog/CDON/Boi/BOC underwent independent duplications in bilaterians, and, as a result, the genes Ihog and Boi appeared in drosophilids and CDON and BOC in vertebrates, while the spiralian animals (e.g., annelids) retained the single ancestral gene *Fibbc* (Father of Ihog/Boi/BOC/CDON). 

### 3.4. Hedgehog Signaling Complex (HSC)

The *Kif27* gene was found in the transcriptomes of both annelids, while sequences homologous to *Kif7* and *Cos2* were not present. At the same time, an NCBI search for Spiralia genes produced the same result. Analysis of the amino acid sequences of Pdum-Kif27 and Pele-Kif27 showed the presence of conserved functional domains among which Kinesin motor domain and microtubule interaction sites were found, which could indicate the retained function of these proteins. Moreover, the phylogenetic tree demonstrated that spiralian Kif27 and KIF7 and KIF27 of deuterostomes fell into one clade, while Cos2 of arthropods branched into a separate clade. Our results are remarkable, since it is known that neither KIF7 nor KIF27 in vertebrates bind to Smo, though the participation of KIF7 in Hh signaling has been demonstrated [97]. Based on our phylogenetic analysis, we suggest that an ancestral gene homologous to *Kif27*, which participated in Hh signaling without interaction with Smo, existed in Urbilateria. In the Ecdysozoa lineage, this gene evolved into Cos2 and acquired the capability to interact with Smo. In Tetrapoda, this ancestral homolog of *KIF27* duplicated into *KIF7* and *KIF27* [97]. As a result, *KIF7* retained the ability to participate in Hh signaling while *KIF27* lost it. In annelids, the ancestral homolog of *Kif27* may be involved in Hh signaling but it does not seem to interact with Smo, since the Smo CTD has no Cos2-binding sites. 

In the *P. dumerilii* and *P. elegans* transcriptomes, we found other components of HSC: Fu/Stk36, Sufu, and PKA, CK1, and GSK3b kinases. Analysis of the sequences suggested that *Pygospio* and *Platynereis* Fu/Stk36 and Sufu proteins are more like their murine orthologs than those of *Drosophila*, which implies common biochemical activity of the annelid and mammalian paralogs. Moreover, according to previous studies on planarians (Lophotrochozoa), Sufu plays an important role in Hh signaling in these animals [31,32]. Thus, it is possible that the Smo-Sufu-Gli axis is highly conserved in the Hh pathway and is retained in different animals. At the same time, the signal transduction from Smo to SuFu-Gli in spiralian animals seems to demonstrate a similarity with vertebrates, as demonstrated here in annelids, and the role of *KIF7* stays with its homolog *Kif27* (Figure 11). 

### 3.5. Signaling Components That Act in Connection with the Primary Cilium

In comparison with *Drosophila*, there are multiple additional components that act between SMO and GLI in mammals. In vertebrates, the primary cilium plays an important role in Hh signal transduction [9,92,102], which consists of IFT proteins (Intraflagellar transport), Kinesin-II, Dynein-2, BBSome proteins and their regulators, transition zone proteins, EVC zone proteins, ciliary membrane proteins, CPLANE complex proteins, and centrosomal proteins (a). Interactions of these proteins are tangled, and several studies are devoted to this topic [109,110,111,112,113,114,115,116,117,118,119,120,121,122,123,124,125,126,127,128,129,130,131,132]. 

It is known that ciliated cells are present in *Drosophila* in the form of type I sensory neurons, which develop at the middle stage of embryogenesis, and also in spermatogenic cells [133]. In *Drosophila*, the Hh signaling cascade significantly depends on Cos2 as a mediator with several molecular functions, while the mammals use a more complicated multistep mechanism of interaction between SMO and GLI, realized through the cilium. Moreover, it was shown that experimental disruption of the primary cilium in fishes, frogs, and other vertebrates created signaling dysfunction, which could indicate the widespread connection of Hh pathway with the cilium in different vertebrate species [93]. In contrast, the knockout of ciliary components in *D. melanogaster* did not result in disruption of Hh signaling [8,134].

In the transcriptomes of both annelids, we found multiple genes for which the role in vertebrate Hh signaling regulation has been studied (Table 1, Appendix A). Despite the fact that we did not find the single ciliary components (EVC and TCTN3) in the studied transcriptomes, all other components were found for at least one annelid (Table 1). Moreover, we found a gene from the ARL13 family (ARL13B). ARL13/ARL13B proteins have a conserved role in signal transduction inside the cilium, and they are lost in species lacking cilia [93].

When comparing the genes which participate in the Hh pathway in the three bilaterian clades, it became obvious that the absence of multiple ciliary proteins in *Drosophila* (according to the NCBI database) correlates with the independence of Hh signaling from the primary cilium in fruit fly development (Table 1). It is also worth mentioning that the set of ciliary genes which we found in *P. dumerilii*, *P. elegans*, and other spiralian animals is much more complete than that of *Drosophila*, and obviously demonstrates more similarity with vertebrate animals. Moreover, the analysis of specific functional domains in the protein sequences showed their conservatism, which may indicate their similar functions. Still, the high similarity of the ciliary gene sets in vertebrates and annelids does not necessarily indicate the common regulation of Hh signaling. In particular, we did not manage to find the highly conserved binding site of Smo to BBSome components, which is present in vertebrates [84,135]. 

According to previously published data, an obvious connection of the cilium to Hh signaling can be detected in echinoderms [93,136,137]. Genome and proteome analyses have also shown the presence of multiple signaling components in the sea urchin *S. purpuratus* that were previously described for vertebrates [104,138]. This indicates that the connection of Hh signaling with the cilium is characteristic not only of vertebrates, but probably of Deuterostomia as a whole. 

When the cilium was incorporated into Hh signaling during the evolution of Bilateria remains an open question. The lack of reliable experimental models among the basal Ecdysozoa and spiral protostomians prevents us from drawing an unambiguous conclusion. In the only experimental work performed on the flatworm *S. mediterranea*, the functional significance of the Fused, KIF27, and Iguana/DZIP1 genes for cilium formation was shown [32]. Based on this research, we can assume that there is an ancestral connection between the Hh pathway and the cilium [1,32]. Surprisingly, interference with the above-mentioned genes does not lead to the disruption of the Hh pathway in planaria. Considering this fact, and the annelid data presented in this research, we propose alternative evolutionary scenarios.

The first scenario (Figure 12, upper scheme) suggests that the last common ancestor of Bilateria (Urbilateria) used the primary cilium for regulation of Hh signaling. In this context, Deuterostomia retained the full form of this connection. The arthropods (*Drosophila*) lost the connection with the cilium on the level of Smo and BBSome interaction in most cells, as did spiralians (planaria and probably annelids), which excluded the motor ciliary protein Kif27 from the Hh pathway. 

The second scenario (Figure 12, middle scheme) assumes the independent integration of the cilium into the Hh pathway in Deuterostomia and Ecdysozoa (on the level of separate neurons). Here, spiralian animals use the ancestral variant of cilium-independent Hh signaling. This means that annelids do not use the cilium for the Hh pathway, despite the availability of a rich repertoire of ciliary proteins. 

The third scenario (lower scheme) proposes the initial connection of the primary cilium with Hh signaling and further clade-specific divergence of the pathway mechanisms. This divergence occurred in the three evolutionary lineages to which certain model animals belong (*C. elegans*, *D. melanogaster*, *S. mediterranea*). In terms of this hypothesis, the annelids could have inherited the ancestral state of Hh signaling, which was initially associated with the cilium (Figure 12).

According to our results, we are not able to conclusively argue for one of the scenarios, since functional tests are needed to prove the involvement of the primary cilium in annelid Hh signaling. 

## 4. Materials and Methods

### 4.1. RNA Isolation

For the gene search, two types of transcriptome were used. The regenerative transcriptomes were obtained from annelids at different stages of regeneration. The worms were cut into two pieces: *P. dumerilii* of around 30 segments long were cut approximately in the middle of the body, and *P. elegans* of around 1.5 cm long were cut after the 20th body segment. For the experiment, we took 25 worms of each species. The worms were allowed to regenerate at 18 °C in seawater in separate Petri dishes for different time periods: 0 hpa (hours postamputation), 4 hpa, 12 hpa, 24 hpa, 48 hpa, and 4 dpa (days postamputation). After that, the newly grown parts, if any, were cut together with 1 or 2 old segments and total RNA was purified out of the fragments using the Quick-RNA miniprep Kit (Zymo Research, Irvine, CA, USA). We repeated the regeneration experiment twice and performed the RNA extraction twice to reproduce the results. Another set of transcriptomes was obtained from juvenile worms. The worms (10 used in the experiment) were cut into 12 pieces and total RNA was purified from each piece using the Direct-zol RNA kit (Zymo Research). The experiment was repeated twice. The quality of the purified RNA was controlled using gel electrophoresis. 

### 4.2. cDNA Library Preparation, Sequencing, and De Novo Transcriptome Assembly

The cDNA libraries were synthesized using the NEBNext Ultra II Directional RNA library Prep kit for Illumina (New England Biolabs, Ipswich, MA, USA). In total, 24 cDNA libraries were sequenced for each species, which included two sets per experiment. The sequencing was performed using an Illumina HiSeq2500 (St. Petersburg State University resource center BioBank) and Illumina NovaSeq 6000 (Evrogen, Moscow, Russia). The processing of paired-end reads included primary quality control using FastQC [139], the correction of sequencing mistakes using Karect [140], the clipping of low-quality and adapter sequences using Trimmomatic [141], and a search for contamination using the *Homo sapiens* reference transcriptome (GENCODE; GRCh38). De novo assembly was performed via Trinity (k-mer = 25) [142]. For decontamination, the MCSC tool was used [45,143]. BioProject has been deposited at NCBI under accession PRJNA901144.

### 4.3. Molecular Characteristics, Sequence Alignment, and Phylogenetic Analyses

The Geneious software was used to predict the open reading frames and translate the cDNA sequences to protein sequences [144]. *Platynereis* coding sequences were also checked for their presence in the genome using Unipro UGENE Local BLAST Search ([145], genome data provided by K. N. Mutemi). For the *Platynereis* and *Pygospio* nucleotide sequences used in the study, see the Appendix A. The candidate proteins were analyzed for the presence of functional domains on the HMMER web server; the search was conducted against the PFAM and SUPERFAMILY profile databases [146]. The DeepTMHMM algorithm was used to predict transmembrane helices [147]. The group-based phosphorylation scoring (GPS) algorithm in the Group-Based Prediction System (5.0) was used to analyze and predict potential phosphorylation sites in the carboxy-terminal regions of Smo proteins [148]. PredGPI tool was used to predict the presence of a GPI anchor in Gas1 proteins [149]. Multiple amino acid sequence alignments of proteins of interest were made in the Geneious software using the MAFFT v7.450 algorithm with the scoring matrix BLOSUM45, and then were manually corrected [144,150]. Phylogenetic analyses were performed using a parallelized version of MrBayes v3.2.7a with four independent runs, sampled every 100 generations with four chains [151]. For each nexus alignment file, a specific model of protein evolution was selected via ProtTest v3.4.2 [152]. Each analysis ran until the average standard deviation of split frequencies between runs was less than 0.01, but the minimum number of generations always was 500,000. A consensus tree and posterior probabilities for each node were calculated from at least the final 3750 generations of each run (minimum 15,000 total trees). Final trees were visualized using FigTree v1.4.4 software (available at http://tree.bio.ed.ac.uk/software/figtree/, accessed on 25 May 2022). Nexus alignment files can be found in the Appendix A.

## 5. Conclusions

We detected a similar set of Hh signaling components in two distantly related annelid species. This indicates the need for and conservatism of the Hh pathway in their common ancestor that lived ~500 mya. The set of core signaling genes, which is similar to that of bilaterians from other evolutionary branches, unites annelids with the rest of Nephrozoa. Nevertheless, the mechanisms of signal transduction are different. Hh signaling in annelids combines features that are present in both Ecdysozoa and Deuterostomia. In particular, the Smo CTDs of annelids possess phosphorylation sites that can be found in vertebrates as well as in *Drosophila*. Annelids use Kif27 as a signal transductor which by all appearances does not directly interact with the Smo receptor. This mechanism is a feature of vertebrates, while the *Drosophila* homolog of Kif27, Cos2, interacts with the Smo CTD. The presence in annelids of the nearly complete set of genes connected to the primary cilium indicates the probable localization of Hh signaling in this compartment, which is a specific feature of vertebrates as well. Nevertheless, we did not find the BBSome-binding site in the intracellular Smo terminus, which is necessary for Smo trafficking inside the cilium. This indicates a possible difference in signaling realization inside the cell between vertebrates and annelids. 

Noteworthy, we found the *Fibbc* gene (Father of Ihog/Boi/BOC/CDON) in the annelids’ transcriptomes, which encodes a Hh ligand co-receptor and seems to be ancestral for Ecdysozoa and Deuterostomia. 

In summary, our transcriptomic surveys indicate greater similarity of the Hh pathway in annelids to that of Deuterostomia than to Ecdysozoa, according to gene repertoire and protein domain composition. At the same time, it is not unthinkable that spiralian animals demonstrate the specific features of the Hh pathway. These features may possess both ancestral and clade-specific traits. In order to choose the most appropriate evolutionary scenario, more model animals and functional analyses of Hh pathway mechanisms are needed.

## Figures and Tables

**Figure 1 ijms-23-14312-f001:**
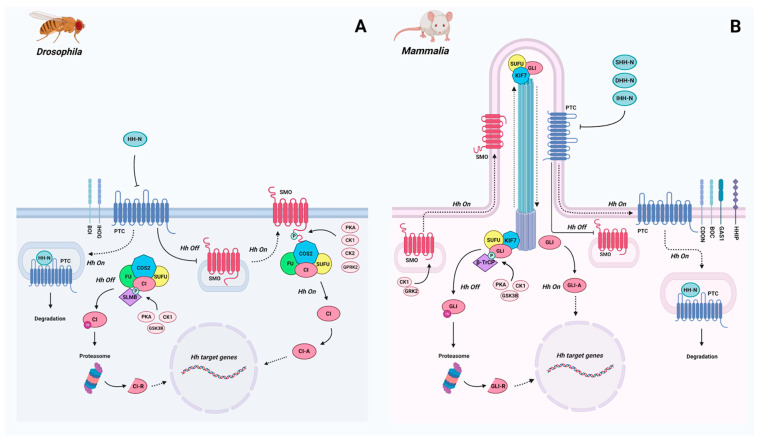
The scheme of the Hedgehog pathway in *Drosophila* (**A**) and mammals (**B**). Hh Off, absence of the ligand; Hh On, presence of the ligand. The dashed lines show the movement of signaling components inside the cell. Arrows indicate the protein interactions and modifications. The Hh ligand is secreted by Hh-producing cells with the help of Disp and interacts with the Ptc receptor on recipient cells. This interaction leads to the removal of the inhibition effect from the other receptor Smo. Active Smo provides signal transduction inside the cell, preventing the cleavage of the transcription factor Gli/Ci. Full-sized Gli/Ci goes to the nucleus and acts as a gene activator [1,12,13,14]. For details, see the Results section.

**Figure 2 ijms-23-14312-f002:**
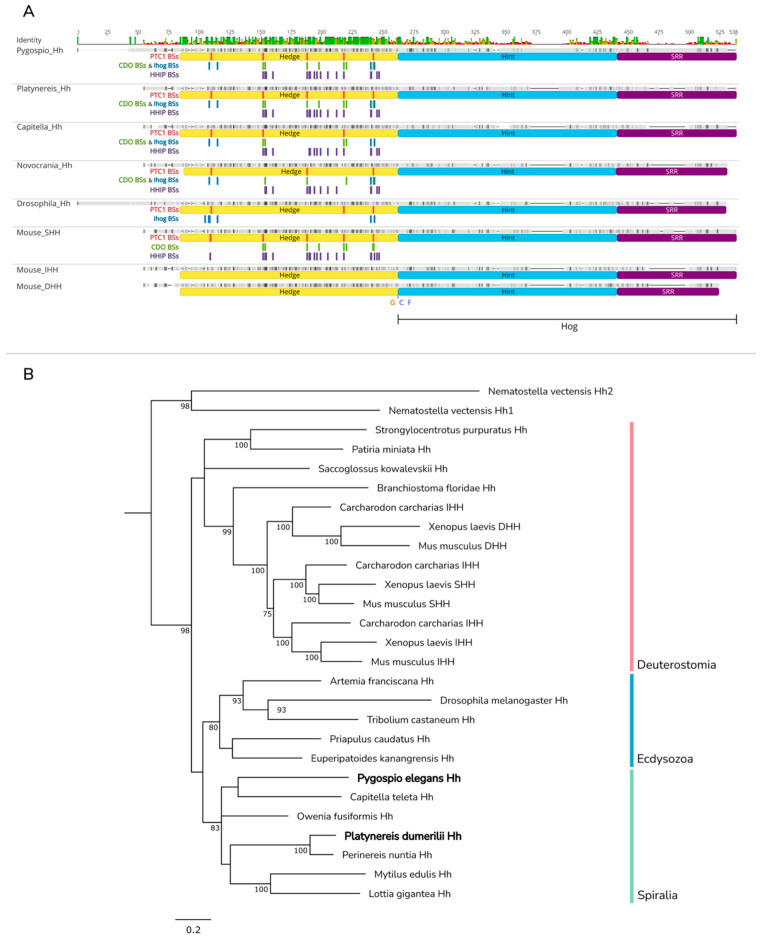
Multiple sequence alignment and phylogenetic analysis of Hh proteins. (**A**) All Hh proteins include an N-terminal Hedge domain and a C-terminal Hog domain. Comparison of Hedge domains from spiralians, *Drosophila*, and mouse revealed that *Pygospio* and *Platynereis* Hh ligands have the sites that are essential for interaction with Patched (indicated by red bars), Ihog (indicated by blue bars), CDO (indicated by green bars), and HHIP (indicated by violet bars). *Pygospio* and *Platynereis* Hedge domains are more similar to murine Hh-Hedge (71.9% and 81.9% similarity, respectively) than to *Drosophila* Hh-Hedge (58.9% and 73.5% similarity, respectively). SRR, sterol-recognition region; PTC1 BSs, Patched-binding sites; CDO BSs, CDO-binding sites; IHOG BSs, IHOG-binding sites; HHIP BSs, HHIP-binding sites. (**B**) Bayesian phylogenetic analysis was performed on Hh protein sequences including Hedge and Hog domains. Hh1 and Hh2 from *Nematostella* were used as an outgroup, as later it was shown that Nv-Hh1 and Nv-Hh2 form a sister group to each other and together are sisters to bilaterian Hh proteins [5]. Numbers near branch nodes indicate Bayesian posterior probabilities. Bayesian posterior probability values less than 75 are not shown.

**Figure 3 ijms-23-14312-f003:**
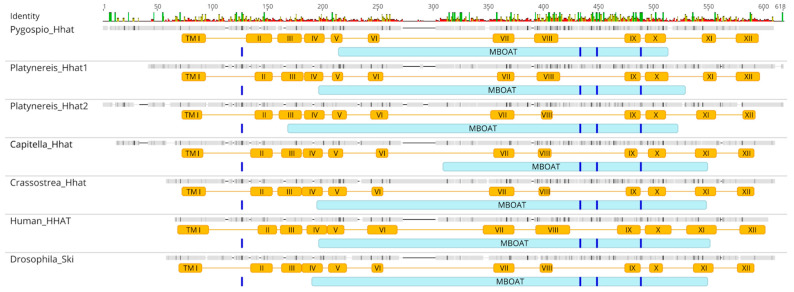
Multiple sequence alignment of Skinny/Hhat proteins. Skinny hedgehog and human HHAT contain 12 transmembrane helices (TM) and an MBOAT domain, which lies in the TM4–TM11 portion of the protein. In human HHAT, mutations at four sites (Glu59, Cys324, Asp339, and His379) reduce protein activity, and these residues are conserved between fruit fly and vertebrates [61]. Spiralian Hhat proteins share a similar domain structure and include conserved sites (indicated by blue bars) essential for Hhat activity. TM (I–XII), transmembrane domains.

**Figure 4 ijms-23-14312-f004:**
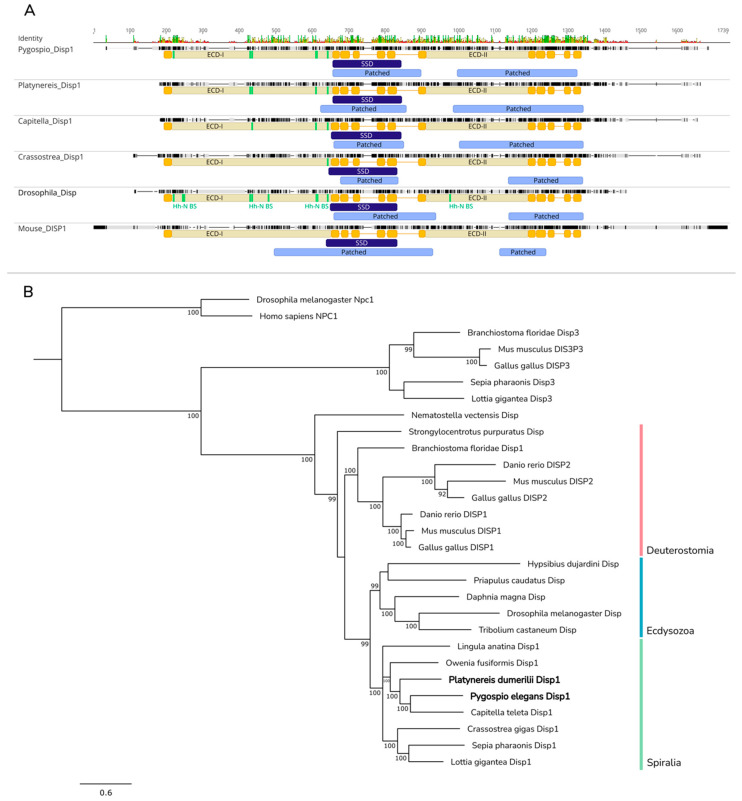
Multiple sequence alignment and phylogenetic analysis of Disp proteins. (**A**) Pdum-Disp1 and Pele-Disp1 share a similar structure with Ptc proteins, comprising 12 transmembrane helices, 2 extracellular domains (ECDs), and conserved Patched domains. As in Ptc, TMs 2–6 form a sterol-sensing domain. Fifteen amino acid residues of *Drosophila* Disp ECDs are involved in Hh-N ligand binding (indicated by green bars). Of these, only relatively few residues are conserved between fruit fly and spiralians (indicated by green bars). Patched, Patched domain (Pfam: PF02460); ECD-I and ECD-II, extracellular domains; TM, transmembrane domain; SSD, sterol-sensing domain; Hh-N BS, Hh-N-binding site. (**B**) Analysis was performed on amino acid sequences localized between terminal transmembrane helices. Pele-Disp1 and *Capitella* Disp formed a sister clade to Pdum-Disp1, and *Owenia* Disp was a basal annelid branch. Putative orthologs of Disp3 proteins were not found in the examined *Platynereis* and *Pygospio* transcriptomes. Numbers near branch nodes indicate Bayesian posterior probabilities. Bayesian posterior probability values less than 75 are not shown.

**Figure 5 ijms-23-14312-f005:**
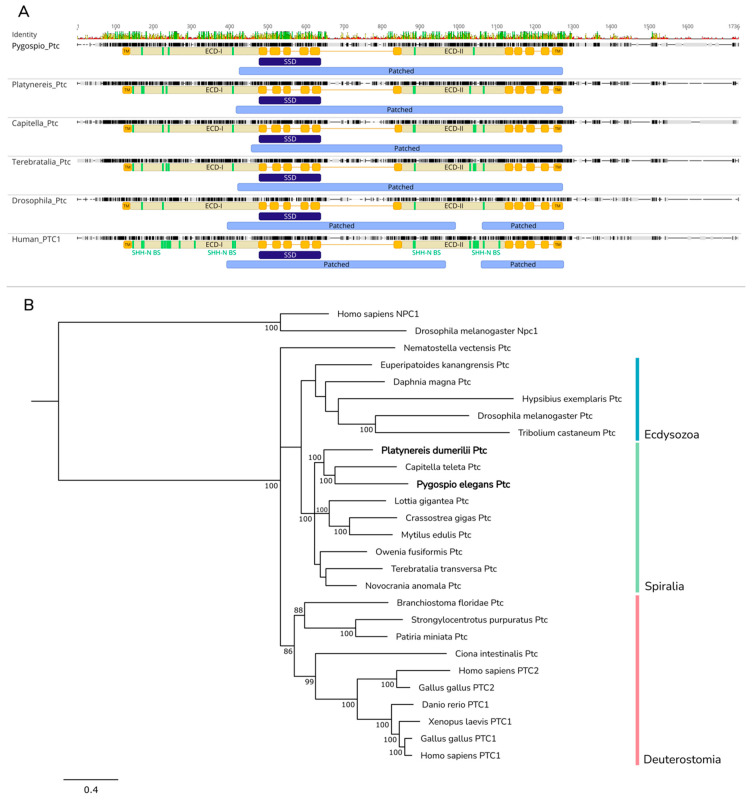
Multiple sequence alignment and phylogenetic analysis of Ptc proteins. (**A**) Pele-Ptc and Pdum-Ptc share a similar structure with Ptc from fruit fly and PTC1 from human, possessing a conserved Patched domain, 12 transmembrane helices (TM), and 2 extracellular domains (ECDs). TMs 2–6 form a sterol-sensing domain. Twenty-one amino acid residues of human ECDs are involved in Hh-N ligand binding (indicated by green bars). Of these, few residues are conserved between human and annelids (indicated by green bars). Patched, Patched domain (Pfam: PF02460); ECD-I and ECD-II, extracellular domains; TM, transmembrane domain; SSD, sterol-sensing domain; Hh-N BS, Hh-N-binding site. (**B**) Bayesian phylogenetic analysis was performed on amino acid sequences containing all 12 TMs, including the conserved Patched domain. Niemann–Pick C1 proteins from fruit fly and human were used as an outgroup. Numbers near branch nodes indicate Bayesian posterior probabilities. Bayesian posterior probability values less than 75 are not shown.

**Figure 6 ijms-23-14312-f006:**
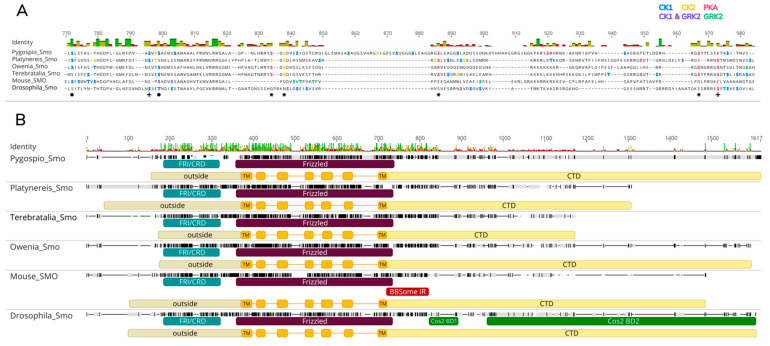
Analysis of Smo phosphorylation sites and domain composition. (**A**) Alignment of PKA, CK1, CK2, and GRK2 phosphorylation sites in Smo proteins. The prediction results suggest that Pele-Smo and Pdum-Smo C-terminal tails may be the phosphorylated substrates of PKA (red), CK1 (blue), CK2 (yellow), and GRK2 (green, common sites with CK1 in violet) kinases. Most phosphorylated residues predicted in annelids are not conserved in fruit fly and mouse. In the studied spiralians, there were three lineage-specific and conserved PKA phosphorylation sites and one CK2 phosphorylation site; these sites are marked with asterisks. Phosphorylation sites that overlapped between annelids and mouse are marked with dots, and sites that overlapped between annelids and fruit fly are marked with plus signs. (**B**) Multiple sequence alignment of Smo proteins. The cysteine-rich and Frizzled domains encompassing the heptahelical TM region were relatively conserved among spiralians, fly, and mammals (59–76% similarity). These parts of the annelids’ Smo proteins shared more identity with the murine protein (45–51% identity) than the fruit fly’s (45–51% identity). The CTD was the most divergent region of the protein (15–53% similarity). FRI/CRD, cysteine-rich domain; TM, transmembrane domain; CTD, C-terminal domain; BBSome IR, the BBSome interacting region; Cos2 BD1, Cos2-binding domain between amino acids 652 and 686; Cos2 BD2, Cos2-binding domain between amino acids 730 and 1035.

**Figure 7 ijms-23-14312-f007:**
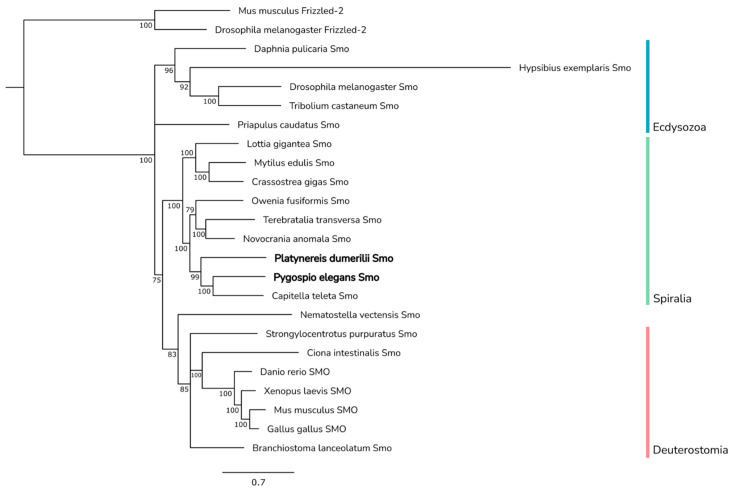
Phylogenetic analysis of Smo proteins. Bayesian phylogenetic analysis was performed using an amino acid alignment of conserved N-terminal regions of metazoan Smo proteins. Pele-Smo and Pdum-Smo, along with *Capitella* Smo, clustered into the Annelida group within the monophyletic Spiralia clade. Smo proteins from deuterostomes and *Nematostella* Smo joined into a separate clade. Ecdysozoan Smo proteins formed a basal branch in the tree. Numbers near branch nodes indicate Bayesian posterior probabilities. Bayesian posterior probability values less than 75 are not shown.

**Figure 8 ijms-23-14312-f008:**
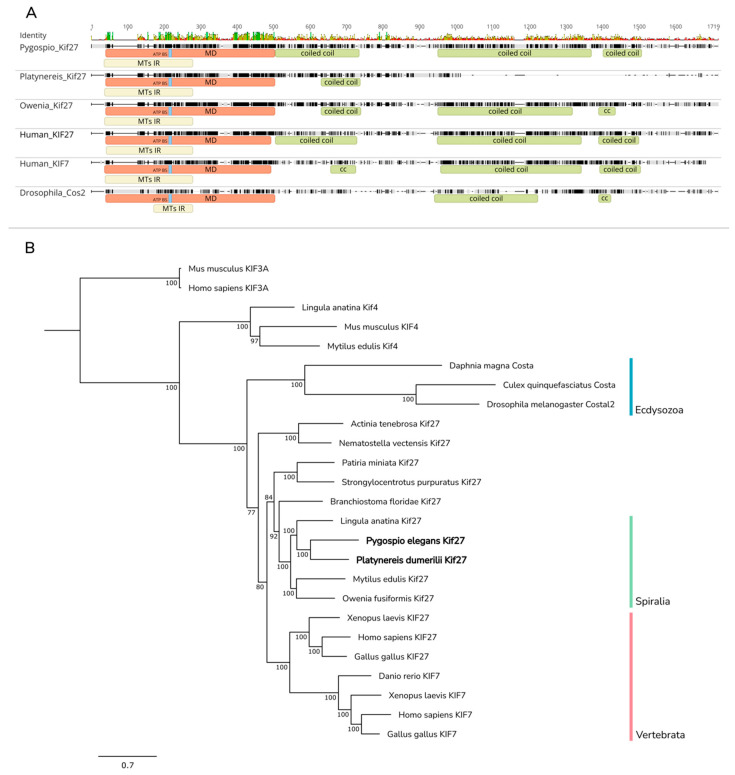
Multiple sequence alignment and phylogenetic analysis of Kif/Cos proteins. (**A**) Kif7 and Kif27 are members of the kinesin superfamily of motor proteins (Kifs) that move along microtubules (MTs) in an ATP-dependent manner. Among metazoan Kif proteins, the structure of human KIF7 and KIF27 is the best studied. Human KIF7 and KIF27 possess an N-terminal globular motor domain that contains an ATP-binding site and MT-interacting region, followed by a stalk domain predicted to form a discontinuous coiled coil. In *Pygospio* and *Platynereis* transcriptomes, we found only Kif27 orthologs that share a similar domain composition with human KIF proteins. MD, motor domain; MT IR, MT-interacting region; ATP BS, ATP-binding site. (**B**) For phylogenetic analysis, full-length amino acid sequence alignment was taken. Homologs of KIF27 are not found in fish lineages, while tetrapods possess both proteins. In *Pygospio*, *Platynereis*, and other spiralian species, we found orthologs of KIF27 and no obvious homologs of KIF7. Vertebrate KIFs fell into one clade with spiralian Kif27, while ecdysozoan Cos proteins formed a basal branch on the tree. Numbers near branch nodes indicate Bayesian posterior probabilities. Bayesian posterior probability values less than 75 are not shown.

**Figure 9 ijms-23-14312-f009:**
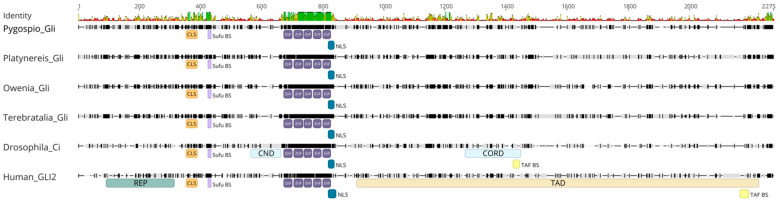
Multiple sequence alignment of Gli proteins. In the N-terminal regions of Pele-Gli and Pdum-Gli, we found a ciliary localization signal and Sufu-binding site. Both domains have previously been identified in mammalian GLIs and Ci and seem to be conserved among Metazoa [93,94]. Pele-Gli and Pdum-Gli also contain a bipartite nuclear localization signal in their Zn-finger DNA-binding domains. Ci, Cubitus interruptus; GLI, glioma-associated oncogene homolog; ZnF, zinc-finger DNA-binding domain; CLS, ciliary localization signal; Sufu BS, Sufu-binding site; NLS, nuclear localization signal; Rep, repressor domain; TAD, transcription activation domain; TAF BS, TAF-binding site; CDN, CORD, Cos2-binding sites.

**Figure 10 ijms-23-14312-f010:**
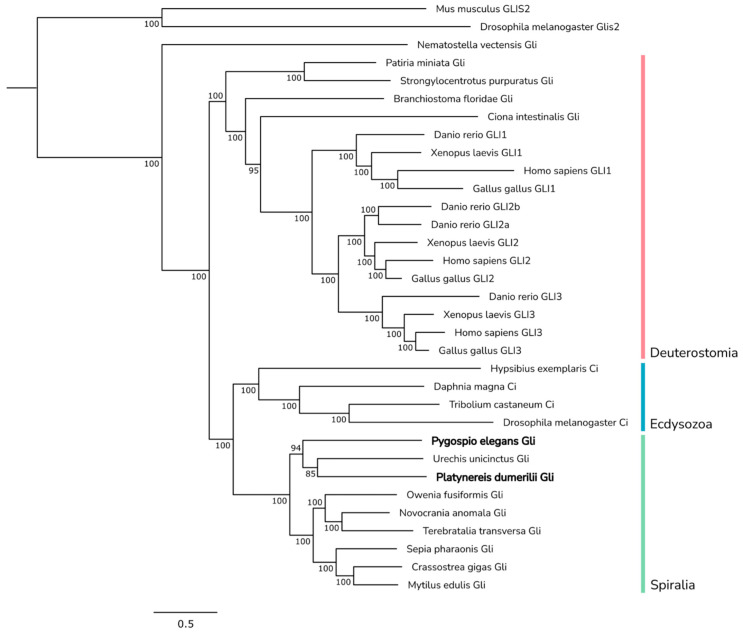
Phylogenetic analysis of Gli proteins. Pele-Gli and Pdum-Gli fell into one clade with Gli from the echiurid *Urechis unicinctus*. GLI-similar 2 (Glis2) proteins from mouse and fruit fly were used as an outgroup. Numbers near branch nodes indicate Bayesian posterior probabilities. Bayesian posterior probability values less than 75 are not shown.

**Figure 11 ijms-23-14312-f011:**
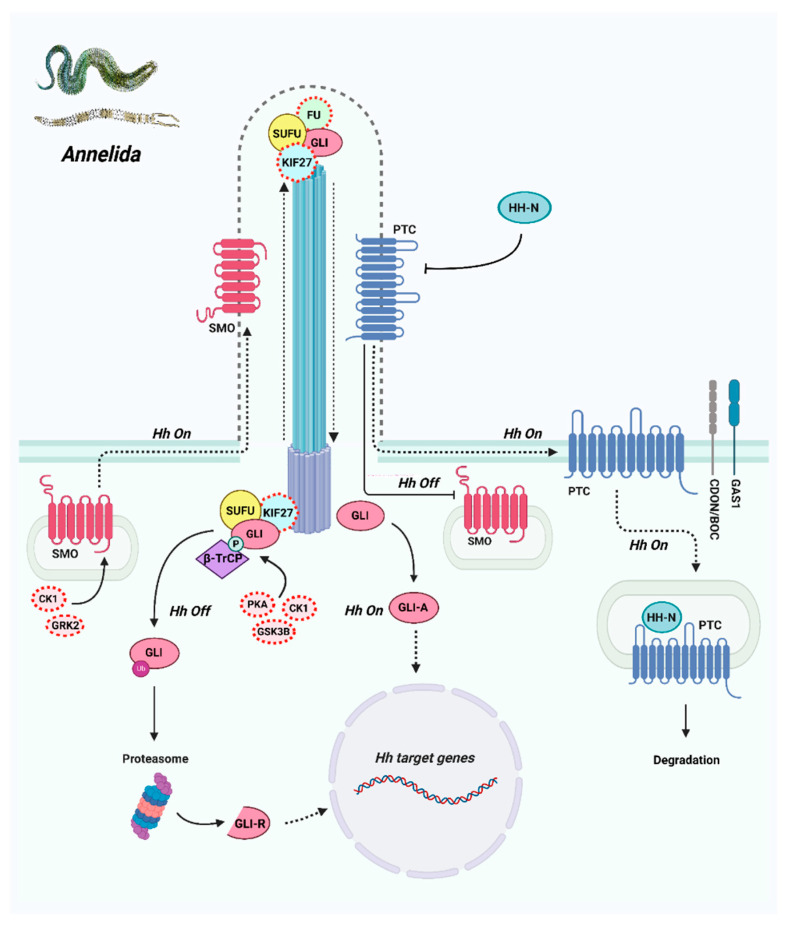
The predicted scheme of the Hedgehog signaling pathway in annelids. The grey dashed line delineates the primary cilium, which may participate in Hh signaling in annelids. The putative signaling participants (compared with these proteins’ functions in other animals) are encircled by red dashed lines.

**Figure 12 ijms-23-14312-f012:**
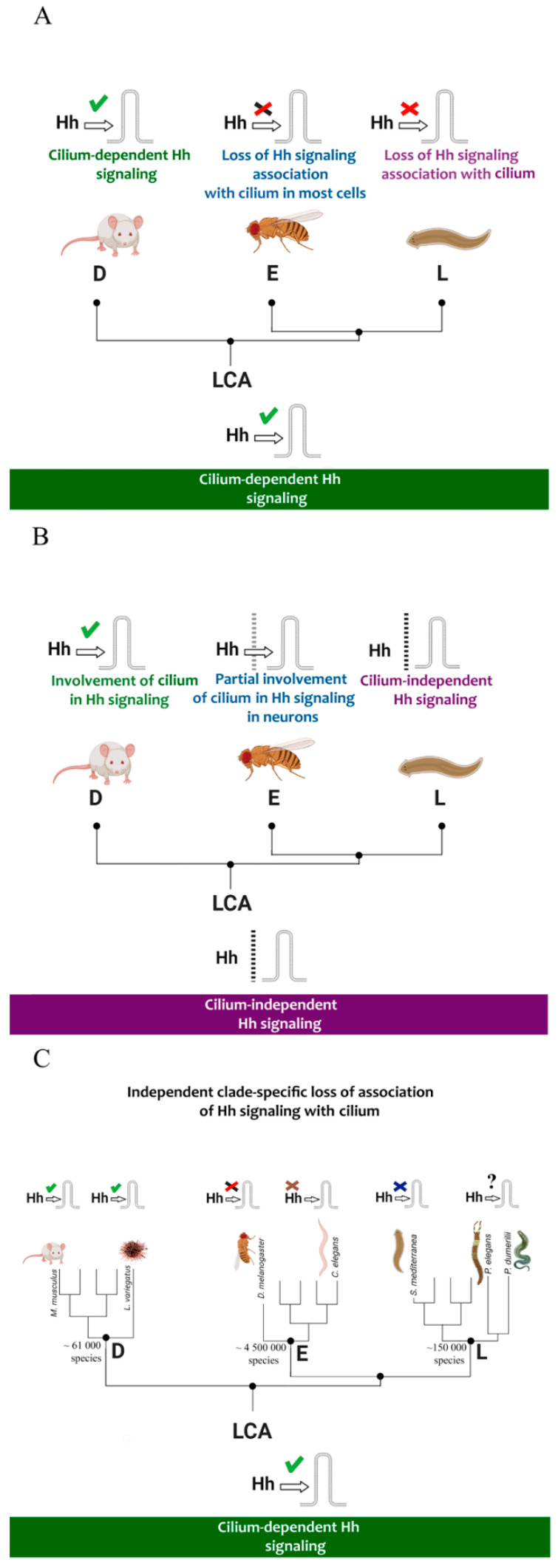
Three possible scenarios for the integration of Hh signaling transduction within the primary cilium in three bilaterian branches: Ecdysozoa, Lophotrochozoa, and Deuterostomia. (**A**) According to the first scenario, cilium-dependent Hh signaling emerged at the level of the last common ancestor of most bilaterian animals (LCA). In the Deuterostomia (D) lineage, this condition was retained, whereas in the Ecdysozoa (E) lineage, the Hh pathway’s association with the cilium was preserved only in particular cell types. In the Lophotrochozoa (L) lineage, this association was lost. (**B**) The second scenario assumes a cilium-independent Hh pathway in the LCA. In this case, D and E independently linked Hh signaling to the cilium. (**C**) In the third scenario, cilium-dependent Hh signaling existed initially, but its high evolutionary lability at the level of intracellular messengers does not allow generalizable conclusions to be drawn for large taxa (D, E, L). Single model animals from a huge number of Ecdysozoa and Lophotrochozoa species could have independently lost their connection between Hh and the cilium at different stages of signal transduction, but this does not reflect the state of the Hh pathway in the Protostomia ancestor.

**Table 1 ijms-23-14312-t001:** Distribution of Hh signaling components in annelids *P. dumerilii* and *P. elegans* (this study), *Drosophila*, and mouse/human. Note that *Drosophila* lacks 26 genes known to participate in vertebrate Hh signaling. +, found in the transcriptome; ++, found both in the transcriptome and genome; –, not found in the transcriptome; – –, not found in the transcriptome or genome; ?, not found only in the *P. elegans* transcriptome; x, not found for this species. For *Drosophila* and mouse/human, amino acid sequence accession numbers are shown, including sequences used in alignments and phylogenetic analyses.

Genes	*P. dumerilii*	*P. elegans*	*D. melanogaster*	*M. musculus/H. sapiens*
Hedgehog	++	+	ABC66186	Mouse SHH NP_033196 Mouse DHH NP_031883 Mouse IHH NP_034674
Patched	++	+	NP_523661	Human PTCH1 AAC50550 Human PTCH2 NP_003729
Smoothened	++	+	NP523443	Mouse ACP30472
Gli/Ci	++	+	AAF59373	Human GLI1 NP_001161081 Human GLI2 NP_001358200 Human GLI3 NP_000159
Dispatched1	++	+	AAF51938	Mouse Q3TDN0
Ski/HHAT	++	+	Q9VZU2	Human Q5VTY9
SCUBE	++	+	x	Mouse Q9JJS0
HHIP	– –	–	x	Human Q96QV1
GAS1	++	+	x	Human AI32683
Ihog/CDON	Fibbc ++	Fibbc +	Ihog Q9VM64	Mouse CDO AAC43031
Boi/BOC			Boi ABW09329	Mouse BOC Q6AZB0
Sufu	++	+	Q9VG38	Mouse Q9Z0P7
Fused/STK36	++	+	P23647	Mouse Q69ZM6
Costal2/KIF7/KIF27	Kif27 ++	Kif27 +	Costal2 O16844	Mouse KIF7 B7ZNG0 Mouse KIF27 Q7M6Z4
Slimb/β-TrCP	++	+	NP_524430	Human NP_378663
IFT25/HSPB11	++	+	x	Human NP_001303864
IFT27	++	+	x	Human NP_001171172
IFT38/CLUAP1	++	?	NP_608470	Human NP_055856
IFT52	++	+	NP_609045	Human NP_001290387
IFT54/TRAF3IP1	++	+	NP_650353	Human NP_056465
IFT56/TTC26	++	+	NP_650486	Human NP_001308671
IFT57	++	+	NP_608792	Human NP_060480
IFT80	++	+	NP_610064	Human NP_065851
IFT81	++	+	x	Human NP_001137251
IFT88/Polaris	++	+	NP_523613	Human NP_001340496
IFT172/Wimple	++	+	NP_647700	Human NP_056477
IFT121/WDR35	++	+	NP_647653	Human NP_065830
IFT122	++	+	NP_648221	Human NP_443711
IFT139/TTC21B	++	+	x	Human NP_079029
IFT140	++	+	NP_608530	Human NP_055529
IFT144/WDR19	++	+	NP_611426	Human NP_079408
KIF3A	++	+	NP_523934	Human NP_001287720
KIF3B	++	+	NP_524029	Human NP_004789
KAP3/KIFAP3	++	+	NP_727512	Human NP_055785
DYNC2H1	++	+	NP_001036369	Human NP_001368
DYNC2LI1	++	+	NP_609289	Human NP_057092
DYNC2I2/WDR34	+	+	x	Human NP_443076
DYNLT2B/TCTEX1D2	++	+	NP_001163579	Human NP_689986
BBS1	++	+	NP_648080	Human NP_078925
BBS2	++	+	x	Human NP_114091
BBS4	++	+	NP_610636	Human NP_149017
BBS5	++	+	NP_649499	Human NP_689597
BBS7	++	+	x	Human NP_789794
BBS8/TTC8	++	+	NP_608524	Human NP_653197
BBS9/PTHB1	++	+	NP_001137727	Human NP_055266
BBIP1/BBS18	++	+	NP_001163568	Human NP_001182233
ARL6/BBS3	++	+	NP_611421	Human NP_001265222
LZTFL1	++	+	x	Human NP_065080
TULP3	++	+	ktub NP_995911	Human NP_003315
TCTN1	++	+	tctn NP_608998	Human NP_001076007
TCTN2	++	+	x	Human NP_079085
TCTN3	– –	–	x	Human NP_056446
MKS1	++	+	NP_572804	Human NP_060247
CC2D2A	++	+	NP_611229	Human NP_001365544
B9D1	++	+	NP_650470	Human NP_056496
TMEM17	++	+	x	Human NP_938017
TMEM107	++	+	x	Human NP_115730
TMEM231	++	+	NP_608928	Human NP_001070884
AHI1/Jouberin	++	+	x	Human NP_001128302
EVC	– –	–	x	Human NP_714928
EVC2	++	+	x	Human NP_667338
IQCE	++	+	x	Human NP_689771
EFCAB7	++	+	x	Human NP_115813
FUZ	++	+	NP_001260250	Human NP_079405
INTU	+	+	NP_788548	Human NP_056508
JBTS17/CPLANE1	++	+	x	Human NP_001371661
RSG1/CPLANE2	++	+	x	Human NP_112169
OFD1	++	+	x	Human NP_003602
C2CD3	++	+	NP_730546	Human NP_001273506
TALPID3	++	?	x	Human NP_001316872
DZIP1/Iguana	++	+	CAC14873	Human NP_055749
GPR161	++	+	x	Human NP_001254538
ARL13B	++	?	x	Human NP_001167621
INPP5e	++	+	NP_648566	Human NP_063945
RAB23	++	+	NP_649574	Human NP_057361
Lophohog	– –	–	x	x

## Data Availability

The data presented in this study are openly available in NCBI at https://www.ncbi.nlm.nih.gov/bioproject/901144, accession number PRJNA901144.

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
