# Peer review of "Gotta Go Slow: Two Evolutionarily Distinct Annelids Retain a Common Hedgehog Pathway Composition, Outlining Its Pan-Bilaterian Core"

_ijms, 2022, doi:10.3390/ijms232214312_

Round 1

Reviewer 1 Report

In this manuscript Platova and collaborators mined the Hedgehog complement members from two annelid species belonging to the two main groups: Sedentaria and Errantia. Through bayesian analyses they identify the orthologs in each species, and make an important contribution in trying to infer the evolution of the pathway within Lophotrochozoans.

I have some major concerns about the interpretations and conclusions regarding a bilaterian complement.

Major:

I interpret this manuscript as the findings of the Hh presence/absence in two annelids, plus comparison with other annelids like Capitella and Owenia, using phylogenetic analyses. To reach to the point of inferring the complements for the bilaterian ancestor is ambitious. The analyses do not include many representatives from Spiralia and Ecdysozoa, only C elegans apart from the arthropods, for example. Even though there are transcriptomic/genomic resources for the main clades, for example onychophora, tardigrades, priapulids, etc. So I would tone down a lot of the main arguments referring to losses or gains in Spiralia or Ecdysozoa, as there is not enough data to support those claims. That includes softing the title referring to the pan-bilaterian core.

Some of the tables and figures seem redundant, and make the manuscript longer than needed.

The authors should try to be more efficient with the use of figures related to a particular point. It would be better to combine the alignment  + phylogenetic tree in a single figure. That will reduce the amount of figures and will use the space more efficiently. E.g., alignment of Hh with tree of Hh. Similar to what the authors did for SF3, SF5 and SF6.

Figure legends are wordy and long. Need to be more concise. They almost take half a page, not really necessary when most figures are single panel figures.

1) Table 1 already provides a good information of what the study found in terms of absence/presence for the two annelids. This is redundant with Figure 15. Either make Figure 15 more concise and put table 1 as supp material, or choose to leave Figure 15 out.

2) I don't find the figure 16 necessary. Figure 16 looks like coming from a text book, too detailed, too broad. Authors could just use a reference for the readers to go and see this diagram on a text book or a review paper.

Minor:

Scientific names need to be in italics, and Drosophila capitilized.

Line 39, Hh should be spelled out the first time used in the main text: Hedgehog (Hh). "is" instead of "was".

Line 40: single component? or single components? . Which protista, if they are closely related to animals, then maybe this is an important point.

Line 48: limb bud "and" instead of "or" neural tube of... of which animal?

Last sentence of first paragraph is confusing as it's written. Maybe something better in the lines of: "In addition, Hh signaling can act via a non-canonical way in roles during cell differentiation and regeneration".

In 19 ocassions the authors should chage the word "conservative" to "conserved".

In the paragraph between line 51 and 56, the last sentence is not needed. Just refered to the figure and table as you describe the pathway in the following paragraph.

Line 62: figure legend, unclear sentence "and its C-terminal activator domain partially depredates through ubiquitin-proteasome pathway." Do the authors mean to say that the domain partially gets DEGRATED through the ubiquitin-proteosome pathway?

Line 93: remove "obviously"

Line 125: "(e.g., Platynereis)" instead of (Platynereis)

Line 129: maybe "important" instead of "perspective"

Phrases starting 133 need some rephrasing. Remove "It is important that these two branches split from the common ancestor a long time ago." , and just state how long ago they branched away from each other.

Paragraph between 143 and 149 is not needed. Authors already stated the position in the annelid phylogeny, and the last paragraph states what the goal of the manuscript is with both worms.

Lines 177-201 are a combination of background and discussion of what is found in other animals. Be concise, use the table you have to depict what each gene role is. That way is easier for the reader to go to the table and understand why you are analysing the phylogeny of those genes. Instead, you are using half a page for each section in the results for something that the readers can find on references.

Lines 303-304: I don't understand why you say that scube is not found, and then say that it is found as the homologue. Therefore, it's is present. Just that scube2 is the name for one of the paralogues in vertebrates. Very confusing.

Lines 306-322: same issue as lines 177-201

Lines 381-410: same issue as lines 177-201

Lines 480-527: same issue as lines 177-201

Lines 552-571: same issue as lines 177-201

Line 830: has a word in Russian животных

Authors should acknowledge K. N. Mutemi for providing access to the genome annotation in the acknowledgements section.

Author Response

Dear Reviewer,

We are very grateful for you taking the time to review our paper. We sincerely appreciate all valuable comments and suggestions.

All the comments we received have been taken into account. Out of these suggestions, we would like to note some points.

  1. Some representatives from Ecdysozoa were included in the phylogenetic analyses. However, C. elegans was not included, as it lacks a complete Hh signaling pathway (see Hao et al., 2006a DOI: 10.1186/1471-2164-7-280). Among Spiralia, we could not include representatives from Gastrotricha, Chaetognatha, and Gnathifera, as there is no transcriptomic/genomic data for some genes, or available sequences are too short to be adequately analyzed.  We made several attempts to include into analysis sequences from model flatworm S. mediterranea, but its’ proteins have changed so much that it led to a long branch attraction bias. Considering this and the fact that flatworms exhibit high developmental and genetic diversity within phylum (see Martin-Duran, J. M., & Egger, B. (2012) DOI: 10.1186/2041-9139-3-7; Grohme, M. A., et al. (2018) DOI: 10.1038/nature25473), they were excluded from our phylogenetic analysis.
  2. We have made the text and figure legends more concise. However, we left some background information about Hh signaling in other animals in each section of the Results, as some readers may not be familiar with the signaling. Moreover, it seems necessary for us to compare our data with current findings for model and non-model organisms.
  3. Figure 16 has been moved to the Supplementary material section. We decided not to leave either Figure 15 or Table 1 out, as the table summarizes all annelids’ genes found in this study, while the figure illustrates our hypothesis concerning how discovered proteins may be involved in annelids’ Hh signaling.
  4. For some proteins, multiple alignments and phylogenetic trees were combined in a single figure. We did not unite figures for Smo and Gli proteins because their alignments have details that are hard to read on a merged figure.

We believe that your suggestions helped us to improve the quality of the manuscript.

Sincerely,

The authors

Reviewer 2 Report

Manuscript ID: ijms-1975099

Title: Gotta go slow: two evolutionary distinct annelids retained a common composition of Hedgehog pathway, outlining its pan-bilaterian core.

Review:

The authors have submitted a comprehensive and well-written manuscript describing components of the Hedgehog signaling pathway in two marine annelids, with hypotheses about signaling in the Lophotrochozoa and Bilaterian ancestor. In doing so, the authors review extensively what is known about the Hedgehog signaling pathway in Drosophila and Mammals. The paper is very long, partly because of its review-like nature. I am not sure if such reviews are typical of IJMS.

Major comments:

I found it difficult to find the main aims in the Introduction, which could be due to the interruption of text by the figures and large Table. The signaling pathway known from Drosophila and mammals is described in detail already here and this information is repeated in the Results in more detail at the beginning of each section before presenting the results from the studied annelids. I think the authors should choose one or the other way of presenting what is known about signaling already (either only in the introduction; or only piece by piece with the results) but not both. It was also a bit confusing to see Table 1 in the Introduction, which lists results from Platynereis and Pygospio. This gave the impression that the information was already known (I wondered if it was background material or results).

The presentation of known understanding of the signaling pathway is of course focused on Drosophila and mammals, but I was expecting more reference and review of what is known from the Lophotrochozoa in the Introduction and more justification of why addition of data from two annelids is important. E.g. on page 5 there is mentioned “Multiple investigations” (but no citation!) and mention of a lack of studies on Lophotrochozoa (but again no citation of what is known). Later, on the same page, the annelids are described as “perfect models” and “perspective model” but no clear justification of why annelids- what about other Lophotrochoza (particularly basal ones)? Also later in the Discussion, would relating the results also to what is known from other Lophotrochozoa be helpful? E.g. on page 21 there is reference to “previous studies of protostomians (no citation! Please add them). I wonder, for example, if the suggestion (page 22) “the common bilaterian ancestor only possessed one copy of each core gene” has already been made previously by others studying other Lophotrochozoa.

The Introduction and interpretations of the results would benefit from a brief mention of the evolutionary relationships of the Bilateria. I think this would be a critical addition to make. Current phylogenomic analyses indicate that Deuterostomia is the basal bilaterian group- see https://www.ncbi.nlm.nih.gov/pmc/articles/PMC7535936/ Several instances in the manuscript imply that Lophotrochozoa are expected to be basal. Also, the hypotheses presented in Figure 17 could be modified to indicate the relationships better. As is, this appears to be a polytomy at the Bilaterian ancestor. Addition of a current understanding of the evolutionary relationships among the groups would help the authors present some expectations for what they should find in the studied annelids. The first paragraph of the Discussion sets out the finding of “presence of almost full complement of genes” in the two annelids, but perhaps this was to be expected? At least, I am not surprised. A more clear statement of the expectations/hypotheses would help.

Can you be more explicit about which of the three hypotheses proposed in Figure 17 has more support given your results and what is known about the relationships of the different groups?

Materials and methods: Please specify the sampling information. How many individuals were included in RNA extraction in each regeneration time point (only one or multiple)? How many juveniles were cut to 12 pieces? Just one? Or were pieces from multiple worms combined?

How many libraries were made? Was any pooling done? Please proved the accession number for the raw data.

Minor comments:

Title: I do not think the “Gotta go slow:” part of the title is needed. I am not sure what it is referring to and, besides, “Gotta” is not proper written English. I would remove this phrase from the title.

Paragraph structure: There are several instances in the manuscript where a paragraph is composed of only one or two sentences, which makes reading somewhat awkward and less fluent. I recommend incorporating these sentences in other paragraphs. For example: First paragraph of the Introduction; Last 3 paragraphs of section 3.3. Several on page 17. Paragraph beginning “Noteworthy” in the Conclusion.

Figure Legends: Some figure legends are quite long and include text that is repetitive from the manuscript. I suggest looking at these carefully and being more concise.

Word choice: “remote annelids” might be better described as “distantly related” or “evolutionarily distant”

Page 6: Capitalize “sea” Barents Sea

Page 9: “The sequence homologues to Disp was” Change “was” to “were”

Page 11: Remove “the” in “demonstrates the similarity” and in “possesses the transporting”; Change “the” to “a” in “has the hydrophobic”

Page 17: Capitalize “drosophila”

Page 17: “in vertebrate’s signaling” is awkward. It might be better to re-phrase as “in the signaling of vertebrates”.

Page 18: Be more specific than “many genes”

Page 18: remove or translate the Cyrillic text

Page 21: Change “Hh signaling investigation” to “investigation of Hh signaling”

Page 22: change “Cos2-binding site retains” to “the Cos2 binding site is retained”

Page 23: change “retaining” to “retention”

Page 23 change “in annelid lineage” to “in the annelid lineage”; remove “the from “have the matching” and from “result of the clade”

Page 23: remove or translate the Cyrillic text

Page 24: “Microtubule” should not be capitalized

Page 24: “extremely remarkable” is maybe too much, I suggest “remarkable”

Page 24: Check the sentence: “neither KIF7 nor KIF27 of vertebrates do not bind”. (meaning they bind?; remove the “do not”). The meaning is awkward.

Page 26: Perhaps it is just the placement of this paragraph between the figures and large figure legends, but the connection of this paragraph with the results is not very clear.

Page 28: remove or translate the Cyrillic text

Page 28: the paragraph beginning “In opposition” is confusing and I suggest rewriting it. For example, the first sentence could be re-phrased to be clearer and more concise. The second sentence describes evolutionary events in a very “purpose”-driven way with terms such as “leadership” and “inventing”. These terms should be avoided as they suggest directed change. The next sentence seems to be missing a verb- should “was found” be added?

Author Response

Dear Reviewer,

We are very grateful for your thoughtful review regarding our manuscript and we sincerely appreciate all valuable comments and suggestions.

All the comments received have been taken into account. We have made a considerable effort to make the Introduction and Discussion sections more concise and informative regarding the evolutionary relationships of Bilateria and what is known about Hh signaling from other Lophotrochozoa. We have also tried to make clear statements of our hypotheses.

In addition, we would like to comment on our title. ‘Gotta go slow’ is a paraphrased reference to 'Gotta Go Fast', which is an opening theme of a famous cartoon about Sonic the Hedgehog. Our results imply that Hh signaling in annelids evolves more slowly than in model ecdysozoans (because two evolutionary distinct annelids retained the signaling composition very similar to one in vertebrates), so we decided to use a pun in our title. We hoped that this wordplay would draw more attention to our paper.

We believe that your suggestions helped us to improve the quality of the manuscript.

Sincerely,

The authors
